# X-Distill: Cross-Architecture Vision Distillation Enables Data-Efficient Visuomotor Learning

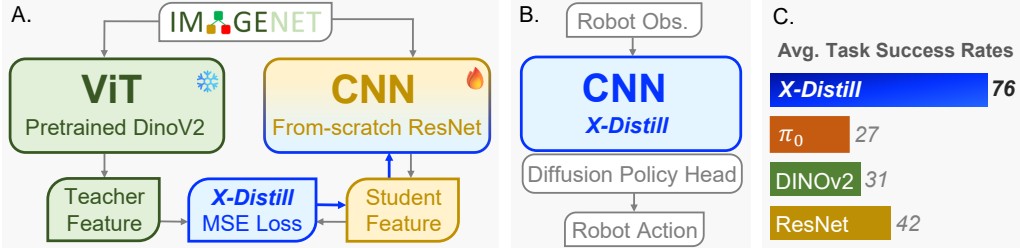

Figure 1: **X-Distill** is a simple yet effective visual encoder enabling data-efficient visuomotor learning. **A.** X-Distill is obtained by cross-architecture knowledge distillation from a large ViT teacher into a compact CNN student on general-purpose image datasets. **B.** Designed for visuomotor policy learning, X-Distill can be jointly fine-tuned end-to-end with a diffusion policy head on robotics-specific datasets. **C.** Given a few ($20 \sim 25$) demonstrations per task, X-Distill significantly outperforms representative counterparts on real-world manipulation tasks, exhibiting its surprising effectiveness.

## ABSTRACT

Visuomotor policies often leverage large pre-trained Vision Transformers (ViTs) for their powerful generalization capabilities. However, their significant data requirements present a major challenge in the data-scarce context of most robotic learning settings, where compact CNNs with strong inductive biases can be more easily optimized. To address this trade-off, we introduce X-Distill, a simple yet highly effective method that synergizes the strengths of both architectures. Our approach involves an offline, cross-architecture knowledge distillation, transferring the rich visual representations of a large, frozen DINOv2 teacher to a compact ResNet-18 student on the general-purpose ImageNet dataset. This distilled encoder, now endowed with powerful visual priors, is then jointly fine-tuned with a diffusion policy head on the target manipulation tasks. Extensive experiments on 34 simulated benchmarks and 5 challenging real-world tasks demonstrate that our method consistently outperforms policies equipped with from-scratch ResNet or fine-tuned DINOv2 encoders. Notably, X-Distill also surpasses 3D encoders that utilize privileged point cloud observations or much larger Vision-Language Models. Our work highlights the efficacy of a simple, well-founded distillation strategy for achieving state-of-the-art performance in data-efficient robotic manipulation.

## 1 INTRODUCTION

Visuomotor policies, exemplified by Diffusion Policy Chi et al. (2023), are promising solutions for generalizable robotic manipulation. As end-to-end approaches, they typically rely on a visual encoder to extract manipulation-centric features from the raw pixels of a scene, followed by a policy head that generates concrete robot actions conditioning on the extracted visual features.

Benefiting from the success of large-scale vision pre-training Caron et al. (2021); Radford et al. (2021), it has become a common practice in recent advances Chi et al. (2024); Lin et al. (2024); Xue et al. (2025); Team et al. (2025) to initialize the visual encoder in a visuomotor policy with off-the-shelf, pre-trained Vision Transformers (ViTs) Dosovitskiy et al. (2021), e.g., CLIP Radford

et al. (2021) or DINOv2 Oquab et al. (2023). These ViT-backend pre-trained models are found to exhibit enhanced generalization capabilities compared to Convolutional Neural Network (CNN) counterparts lacking open-world semantic knowledge, e.g., a ResNet He et al. (2016) trained from scratch.

However, lacking the strong intrinsic inductive biases inherent to CNNs, such as locality and translation equivariance, ViTs are known to struggle when faced with limited amounts of training data Dosovitskiy et al. (2021); Touvron et al. (2021). This issue becomes prominent and inevitable in the context of robot learning, where the dataset size is significantly smaller than in computer vision. Despite the recent trend among embodied AI startups to train policies with hundreds of *hours* of high-quality data produced by a data collection factory Black et al. (2024); Bu et al. (2025); Jiang et al. (2025), most researchers in academia typically collect data by hand, thus favoring data-efficient policies that perform well under a dataset size constraint of tens to a few hundred manipulation *trajectories*.

In this work, we find that simple advances to the visual encoder can yield higher-performing and more data-efficient visuomotor policies. More specifically, we design a cross-architecture vision distillation mechanism, or **X-Distill** in short, which attempts to combine the merits of both the open-world semantic generalization capabilities of pre-trained ViT models, and the inductive bias of CNN architectures that facilitate policy optimization under the low-data regime.

On an implementation level, we instantiate X-Distill by selecting DINOv2 (ViT-L/14) as the teacher encoder, a lightweight from-scratch ResNet-18 as the student encoder, and the mean squared error (MSE) between the teacher and student features as the knowledge distillation Hinton et al. (2015) loss. To make the X-Distilled encoder generally effective for diverse tasks, environments, and robot platforms, we choose the general-purpose ImageNet dataset as the distillation corpus, avoiding potential overfitting to any specific robotic scenarios. After X-Distillation, the CNN-backend encoder with ViT pre-training knowledge can be seamlessly integrated into the policy learning pipeline, jointly fine-tuned with the policy head in an end-to-end manner on any robotics-specific datasets.

We validate the effectiveness of X-Distill by conducting experiments on 34 simulated tasks across MetaWorld (Yu et al., 2020), Adroit (Kumar, 2016; Rajeswaran et al., 2017b), and DexArt (Bao et al., 2023) benchmarks, with 10 demonstrations per task. We also design 5 real-world tasks, carefully defining their In-Distribution (ID) and Out-of-Distribution (OOD) conditions and preparing $20 \sim 25$ demonstrations per task. Empirically, we find that Diffusion Policy with X-Distill consistently outperforms counterparts equipped with ResNet from scratch or DINOv2 as the encoder. Additionally, our policy also outperforms 3D Diffusion Policy Ze et al. (2024), which utilizes privileged 3D observation, as well as $\pi_0$ Black et al. (2024), a Vision-Language-Action (VLA) model that adopts a much larger VLM Beyer et al. (2024) as the visual perception encoder. Finally, we present a detailed qualitative analysis of the learned representations, providing insights into how X-Distill achieves superior performance over the baseline methods. ***Please refer to the*** *project website* ***for robot videos.***

## 2 RELATED WORK

### 2.1 VISUAL REPRESENTATION LEARNING

After the dominance of Convolutional Neural Networks (CNNs) He et al. (2016); Simonyan & Zisserman (2014); Tan & Le (2019) in the 2010s, Vision Transformers (ViTs) Dosovitskiy et al. (2021); Touvron et al. (2021); Liu et al. (2021) have gained increasing popularity in the 2020s because of their superior scaling capabilities and impressive representational power when pre-trained on large-scale datasets Caron et al. (2021); He et al. (2020; 2022); Oquab et al. (2023). Despite this trend, CNNs maintain a crucial edge in low-data regimes and continue to see widespread practical deployment. The key reason for this is the strong **inductive bias** inherent in CNNs. The convolutional operator imposes assumptions of locality and spatial weight sharing, which make them remarkably data-efficient. On the other hand, ViTs lack such biases and therefore require exposure to massive datasets to learn fundamental visual concepts. This discrepancy in data requirements keeps CNNs popular in many specialized domains, such as medical diagnostics Shamshad et al. (2023) or manufacturing quality control Liu et al. (2024), where large labeled datasets are often unavailable.

Recent works also explore adapting frozen models via task-specific adapters Sharma et al. (2023); Liu et al. (2023). Due to the lack of public codebases, we evaluate this paradigm using LoRA Hu et al. (2022). Our results (Sec. 4) show that in data-scarce regimes, our cross-architecture distillation significantly outperforms such parameter-efficient fine-tuning (PEFT) methods.

While parameter-efficient fine-tuning (PEFT) using adapters has been proposed Sharma et al. (2023); Liu et al. (2023), our experiments with LoRA Hu et al. (2022) reveal that in extremely data-scarce regimes, cross-architecture distillation yields superior sample efficiency compared to adapting large ViTs.

## 2.2 CROSS-ARCHITECTURE KNOWLEDGE DISTILLATION

Knowledge distillation (KD) has become a cornerstone technique for model compression and knowledge transfer. Most work has focused on distillation between **homologous architectures**, including traditional CNN-to-CNN approaches Hinton et al. (2015) and more modern ViT-to-ViT frameworks designed for efficiency such as TinyViT Wu et al. (2022). For robotics, homologous knowledge distillation was recently explored by Theia (Shang et al., 2024), which fuses knowledge from multiple pre-trained ViTs into a single unified ViT encoder. In comparison, **cross-architecture distillation** is comparatively underexplored. Liu et al. (2022a). A representative work is DeiT Touvron et al. (2021), a CNN-to-ViT distillation where a CNN teacher can stabilize a data-hungry ViT student. By contrast, our work adopts the converse approach: a ViT-to-CNN distillation aiming to combine the inductive bias of a CNN with the powerful semantic understanding of a large-scale pre-trained ViT.

## 2.3 VISUOMOTOR POLICY LEARNING

Visuomotor policy learning is a promising paradigm for robotic manipulation. Representative works in this vein include Diffusion Policy Chi et al. (2023) and related approaches (Pearce et al., 2023; Reuss et al., 2023; Xian et al., 2023; Hu et al., 2024; Sridhar et al., 2024; Prasad et al., 2024), which typically consists of a visual encoder followed by a policy head network. Recently, Vision-Language-Action (VLA) models have been proposed to replace the visual encoder with more capable vision-language models (VLMs) Brohan et al. (2023); Reed et al. (2023); Black et al. (2024); Team et al. (2024); Liu et al. (2025), enabling impressive generalization abilities such as zero-shot skill deployment in unseen homes Intelligence et al. (2025). However, finetuning the VLM requires a substantial amount of training data. State-of-the-art VLAs such as $\pi_0$ Black et al. (2024), AgiBot GO-1 Bu et al. (2025), and Galaxea G0 Jiang et al. (2025) all rely on their embodiment-specific large-scale datasets, measured either in millions by the number of trajectories or in hundreds of hours by the physical on-robot execution time. In this work, we focus on training capable visuomotor policies when a limited amount of training data is available, i.e., using only $\sim 25$ demonstration trajectories per task.

## 3 METHOD

### 3.1 X-DISTILL: A CROSS-ARCHITECTURE DISTILLATION METHOD

As detailed in Algorithm 1a in Appendix A, we employ cross-architecture knowledge distillation to transfer the representational capabilities of a large Vision Transformer (ViT) into a compact CNN with beneficial inductive biases. Crucially, this entire process is conducted exclusively on the general-purpose ImageNet-1K Deng et al. (2009) dataset ($\mathcal{X}$), which contains approximately 1.3 million images depicting a wide variety of real-world objects and scenes. This decoupling of visual feature distillation from the downstream domain-specific datasets makes X-Distill entirely **domain-agnostic**. In other words, the resulting X-Distill encoder is universally suitable for all kinds of robotic manipulation tasks, thus avoiding potential overfitting to any specific environments, camera setups, or robotic embodiments.

**Selection of teacher and student networks.** We select the pre-trained DINOv2 (ViT-L/14) model as our teacher $\mathcal{T}$. With approximately 304M parameters, this large-scale model is used off-the-shelf as a frozen feature extractor, serving as a robust source of semantic and structural visual knowledge. For the student model $\mathcal{S}$, we choose a highly compact ResNet-18 architecture with only

11M parameters. The choice of student network prioritizes not only its computational efficiency with a network parameter size nearly $28\times$ smaller than the teacher, but also its strong inductive biases such as spatial locality that are beneficial for manipulation tasks.

**Domain-agnostic distillation.** The student is trained to replicate the feature outputs of the teacher on ImageNet-1K. For a given input image $x$, we extract the global `[CLS]` token from the DINOv2 teacher, which serves as the target feature vector. The ResNet-18 student architecture is modified with a final linear layer to match the feature dimension of the teacher. The core objective is then to minimize the direct Mean Squared Error (MSE) between these two feature vectors:

$$\mathcal{L}_{\text{KD}} = \mathbb{E}_{x \sim \mathcal{X}} \left[ \|f_{\mathcal{T}}(x) - f_{\mathcal{S}}(x)\|_2^2 \right] \tag{1}$$

where $f_{\mathcal{T}}$ and $f_{\mathcal{S}}$ represent the complete feature extraction processes of the teacher and the dimension-aligned student, respectively. This process results in a ResNet-18 with parameter weights $\mathcal{S}^*$, which encodes the open-world generalization knowledge of the teacher network.

### 3.2 FINETUNING X-DISTILL FOR VISUOMOTOR POLICY LEARNING

Given the powerful initialization provided by X-Distill, we deploy the encoder $\mathcal{S}^*$ for downstream policy learning on a target robotics dataset, as outlined in Algorithm 1b. We use a Diffusion Policy Chi et al. (2023) head, which generates action chunks conditioned on robot observations.

At each timestep, the distilled encoder $\mathcal{S}^*$ processes a history of camera images $x_{t-T_o+1:t}$ into a visual feature vector $z_{\text{img}}$. This vector is concatenated with the robot's proprioceptive state $s_t$ to form a comprehensive conditioning vector, $c = \text{concat}(z_{\text{img}}, s_t)$. This conditioning vector $c$ guides the entire action generation process. During inference, actions are generated by iteratively denoising a random Gaussian tensor, conditioned on this vector $c$.

Crucially, both the distilled encoder $\mathcal{S}^*$ and the diffusion policy head $\pi_\theta$ are jointly trained on robotics-specific datasets. This end-to-end optimization allows the powerful, general-purpose features from the distillation phase to be fine-tuned and specialized for the specific demands of the manipulation task. The entire system is optimized by minimizing the diffusion loss objective:

$$\mathcal{L}_{\text{diff}} = \mathbb{E}_{\mathbf{A}^0, \epsilon, k} \left[ \|\epsilon - \epsilon_\theta(\mathbf{A}^0 + \sigma_k \epsilon | c, k)\|^2 \right], \tag{2}$$

where $\mathbf{A}^0$ denotes the ground-truth actions, $\epsilon \sim \mathcal{N}(0, \mathbf{I})$, and $k$ is sampled from the diffusion steps.

## 4 SIMULATION EXPERIMENTS

### 4.1 SETUP

**Simulation benchmarks.** To thoroughly evaluate the effectiveness of our method, we conduct experiments across a total of 34 tasks from 3 distinct MuJoCo-based robotic manipulation benchmarks. Our evaluation encompasses tasks requiring parallel gripper manipulation from MetaWorld (Yu et al., 2020), dexterous motor skills from Adroit (Kumar, 2016; Rajeswaran et al., 2017b), and articulated object manipulation from DexArt (Bao et al., 2023). Tasks in MetaWorld are categorized into various difficulty levels—*easy*, *medium*, *hard*, and *very hard*—based on Seo et al. (2023). A brief overview of the tasks is provided in Appendix B.

**Expert demonstrations.** 10 trajectories are collected for each simulation task. For MetaWorld, scripted policies are employed. Trajectories in the remaining domains are gathered using agents trained via reinforcement learning (RL): specifically, VRL3 (Wang et al., 2022) is applied for Adroit, while PPO (Schulman et al., 2017) is utilized for the remaining benchmarks. Other training-related hyperparameters can be found in the Appendix C.

**Evaluation Metric.** We report all results averaged over 3 random seeds (0, 1, and 2). For each individual training run, we evaluate the policy on 20 episodes every 200 epochs, and the highest success rate achieved throughout the run is reported for that seed. The final values presented in our tables are the mean of these scores across the 3 seeds.

Table 1: **Averaged success rates on MetaWorld, Adroit and Dexart benchmarks.** PointNet-DP3 is marked in gray because it processes privileged background-cropped 3D point clouds.

| Method | MetaWorld | | | | Adroit (3) | Dexart (2) | Average |
|---|---|---|---|---|---|---|---|
| | (easy 20) | (medium 7) | (hard 1) | (very hard 1) | | | |
| ResNet-scratch | 76.6 | 48.0 | 38.0 | 50.0 | 37.7 | 54.5 | 64.1 |
| DINOv2 | 78.5 | 46.0 | **48.0** | 38.0 | 51.7 | 58.0 | 66.2 |
| Depth-Anything | 68.2 | 29.3 | 42.0 | 43.0 | 40.3 | **66.0** | 56.1 |
| Theia | 50.9 | 13.7 | 0.0 | 38.3 | 8.7 | 24.0 | 36.0 |
| **X-Distill (Ours)** | **93.9** | **88.3** | **48.0** | **88.0** | **68.3** | 63.5 | **87.2** |
| PointNet-DP3 | 90.4 | 70.6 | 14.0 | 72.0 | 40.7 | 85.0 | 84.0 |

## 4.2 PERFORMANCE

**Compared methods.** We compare X-Distilled ResNet-18 (11M) against several visual encoder counterparts with a similar number of parameters, including:

- **ResNet-scratch** (He et al., 2016), ResNet-18 (11M) trained from scratch;
- **DINOv2** (Oquab et al., 2023), ViT-small (21M) pre-trained using large-scale self-supervision;
- **Depth-Anything** (Yang et al., 2024), ViT-small (24.8M) trained for monocular depth estimation;
- **Theia** (Shang et al., 2024), ViT-small (22M) that distills multiple vision foundation models.

Additionally, we also benchmark against **PointNet-DP3** (Ze et al., 2024), a PointNet-based architecture (0.06M) processing privileged background-cropped 3D point cloud observations.

**Main results.** As summarized in Table 1, X-Distill achieves the best overall average performance across all 34 tasks. It consistently outperforms all 2D vision baselines by a significant margin, securing state-of-the-art success rates in most simulation benchmarks. This validates the effectiveness of our distillation strategy for data-scarce visuomotor learning.

Notably, our 2D approach remains highly competitive even in geometrically demanding settings where methods leveraging privileged 3D inputs have a natural advantage. For instance, the DexArt-Toilet task requires the robot to lift the toilet lid from a frontal viewpoint, which is inherently challenging to estimate the depth relationship between the gripper and the object to be manipulated from a single RGB image. Nevertheless, X-Distill still demonstrates decent performance in many of these challenging tasks, showcasing a strong prior in spatial reasoning. More detailed settings and results are available in Appendix B and Figure 6.

Table 2: **Ablation study on MetaWorld benchmarks.** We evaluate the impact of teacher model scale (DINOv2-L vs. S), student architectural bias (CNN vs. ViT), and student model scale.

| Teacher | Student | MW-20 (easy) | MW-7 (medium) | MW-1 (hard) | MW-1 (v. hard) | Average |
|---|---|---|---|---|---|---|
| DINOv2-L | ResNet-18 (11M) | 93.9 | **88.3** | 48.0 | 88.0 | **90.7** |
| | ViT-S-Half (11M) | 72.0 | 25.3 | 2.0 | 40.0 | 57.2 |
| | ConvNeXt (89M) | 91.8 | 77.4 | **50.0** | 83.0 | 86.6 |
| DINOv2-S | ResNet-18 (11M) | **94.3** | 87.3 | 43.0 | **90.0** | 90.6 |

## 4.3 ABLATION STUDIES

We conduct ablation studies to investigate the impact of the teacher network parameter size, as well as the student network architectural bias and parameter size within our X-Distill framework. The ablation results are summarized in Table 2.

**Teacher network parameter size.** We distill **DINOv2-S** (21M) and **DINOv2-L** (304M) teachers into the same ResNet-18 student. No significant difference can be observed between DINOv2-S and DINOv2-L, indicating our X-Distill framework is insensitive to the specific network configurations of a well-pre-trained teacher network. Nevertheless, we use the DINOv2-L teacher for all subsequent experiments to ensure the maximized knowledge quality that the teacher could provide.

**Student network architectural bias.** We distill the same DINOv2-L teacher into a **ResNet-18** (11M) and a customized **ViT-S-Half** (11M) of the same size. The ResNet-18 student substantially

**Move Cube**   **Move Brush**   **Writing**   **Drawer Open**   **Door Close**

Figure 2: **Visualization of configurations for our real-world tasks.** The orange arrow provides a schematic representation of the gripper trajectory as derived from the data. The green regions represent the distribution of object/robot configurations seen during training demonstrations, while the red regions illustrate the novel configurations used for generalization testing.

outperforms its ViT counterpart by 33.5%. This highlights the crucial role of convolutional inductive biases for visuomotor learning in a low-data regime, supporting our primary hypothesis.

**Student network parameter size.** We compare our compact **ResNet-18** (11M) student to a much larger **ConvNeXt** (89M) Liu et al. (2022b) CNN counterpart. Despite its greater capacity, the larger model achieves a slightly degraded success rate by 4.1% on robotics tasks. This confirms our intuition that smaller visual encoders with stronger inductive biases are easier to optimize, thus beneficial for data-efficient policy learning.

## 5 REAL-WORLD EXPERIMENTS

### 5.1 EXPERIMENT SETUP

We conduct all real-world experiments with an X-Arm 6 robotic arm, capture image observations through a web camera at 15Hz, and prepare a small collection of demonstrations ($20 \sim 25$) per task via Meta-Quest VR teleoperation (in Appendix E). We design 5 tabletop manipulation tasks, and carefully define their In-Distribution (ID) and Out-of-Distribution (OOD) object randomization ranges for rigorous and repeatable evaluation. Task execution trajectories as well as ID and OOD ranges are illustrated in Figure 2. Detailed numbers of demonstrations and evaluation trials can be found in Table 3.

More specifically, **Move Cube** requires the robot to pick up an orange cube and place it into a bowl. In addition to testing on OOD cube positions, we also conduct a color generalization (C-Gen) test with unseen yellow and green cubes. **Move Brush** requires the robot to pick up a brush pen with various initial orientational and translational offsets and place it onto a stand. **Writing "AGI"** requires the robot to sequentially write letters "AGI" on a randomly placed piece of paper. We conduct OOD dynamic perturbation trials, where human perturbators randomly drag the paper elsewhere while the robot is writing letters. **Drawer Open** requires the robot to insert its finger into varying initial gaps of the randomly placed drawer, and then open it by sliding outward. **Door Close** requires the robot to close the door from various initial open angles by pushing it inward. More task descriptions are provided in the Appendix F.

### 5.2 EXPERIMENT RESULTS AND ANALYSIS

**Baselines.** We compare our X-Distill encoder against three representative counterparts. The first two are Diffusion Policies equipped with either a ResNet encoder trained from scratch or an off-the-shelf DINOv2 encoder. Both of the two baseline policies, as well as our approach, are trained for 1500 epochs on our task-specific data. Our third baseline is the state-of-the-art Vision-Language-Action (VLA) model, $\pi_0$ Black et al. (2024). Considering its significant computational requirements, we performed supervised fine-tuning (SFT) for $30,000$ steps, following the official recommendations,

Table 3: **A comparison of task execution success rates (%) for real-world tasks**, along with the numbers of demonstrations and evaluation trials.

| Method | Move Cube | | | Move Brush | | Writing "AGI" | | Drawer Open | | Door Close | | Average |
|---|---|---|---|---|---|---|---|---|---|---|---|---|
| | ID | OOD | C-Gen | ID | OOD | ID | OOD | ID | OOD | ID | OOD | |
| # Demos | 20 | 0 | 0 | 24 | 0 | 25 | 0 | 20 | 0 | 20 | 0 | – |
| # Eval Trials | 15 | 5 | 10 | 4 | 8 | 5 | 4 | 5 | 15 | 5 | 10 | – |
| ResNet-scratch | 66.7 | 0.0 | 50.0 | 0.0 | 0.0 | 40.0 | 25.0 | **100.0** | 13.3 | 60.0 | 80.0 | 41.9 |
| DINOv2 | 26.7 | 20.0 | 20.0 | 0.0 | 0.0 | 0.0 | 0.0 | 80.0 | 13.3 | **100.0** | 90.0 | 31.4 |
| $\pi_0$ (SFT) | 0.0 | 0.0 | 0.0 | 25.0 | 0.0 | 0.0 | 0.0 | 80.0 | 33.3 | 80.0 | 90.0 | 26.7 |
| **X-Distill (Ours)** | **93.3** | **40.0** | **70.0** | **75.0** | **25.0** | **100.0** | **100.0** | **100.0** | **53.3** | **100.0** | **100.0** | **75.6** |

which took approximately 20 hours on a single A100 GPU. All methods were trained using the same dataset consisting of the same limited number of demonstrations to ensure a fair comparison.

**Main results.** The quantitative results for real-world experiments are summarized in Table 3. X-Distill demonstrates clear superiority, consistently outperforming all baseline approaches by a large margin and achieving the highest success rates across both ID and OOD evaluation settings. Simply finetuning a large ViT encoder like DINOv2 yields poor performance, confirming the challenge of effectively optimizing large Transformer networks in data-scarce scenarios and underscoring the effectiveness of our cross-architecture distillation.

The performance gap is particularly insightful when comparing against the VLA model, $\pi_0$. While $\pi_0$ shows reasonable success on simpler tasks like **Drawer Open**, it struggles significantly on more complex, high-precision tasks such as **Writing "AGI"**, where its performance drops to zero. This suggests that directly finetuning a large, generalist VLA on small, task-specific datasets is a significant challenge. In contrast, our X-Distill framework effectively bridges this gap by transferring knowledge into a compact, data-efficient architecture, highlighting the importance of matching the model and pre-training strategy to the available data resources.

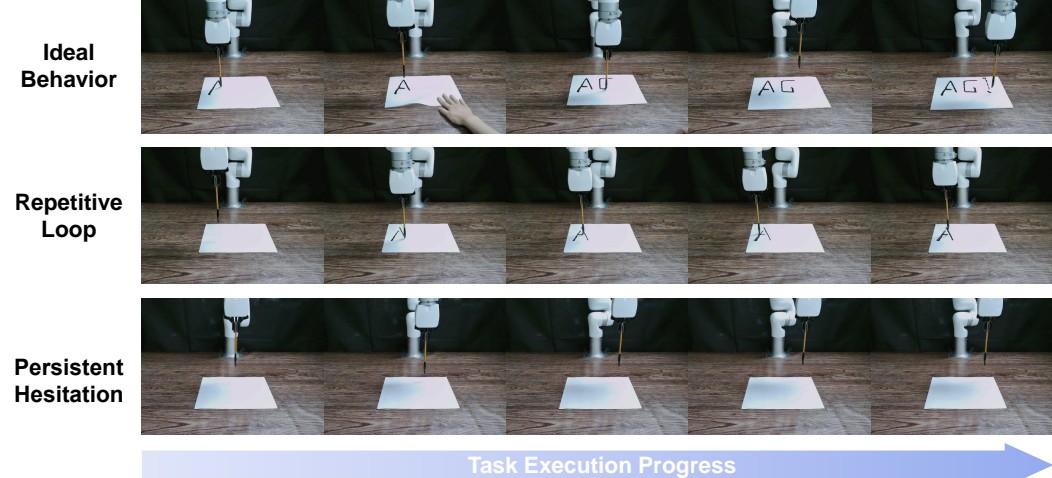

Figure 3: **Representative trajectory types observed in the "Writing AGI" task.** We identify three distinct behaviors: **(1) Ideal Behavior:** Successful and robust execution of all three letters, even under perturbation. **(2) Repetitive Loop:** Perseverative behavior where the policy gets stuck repeatedly writing the first letter 'A'. **(3) Persistent Hesitation:** Dithering motion above the paper without initiating the writing task.

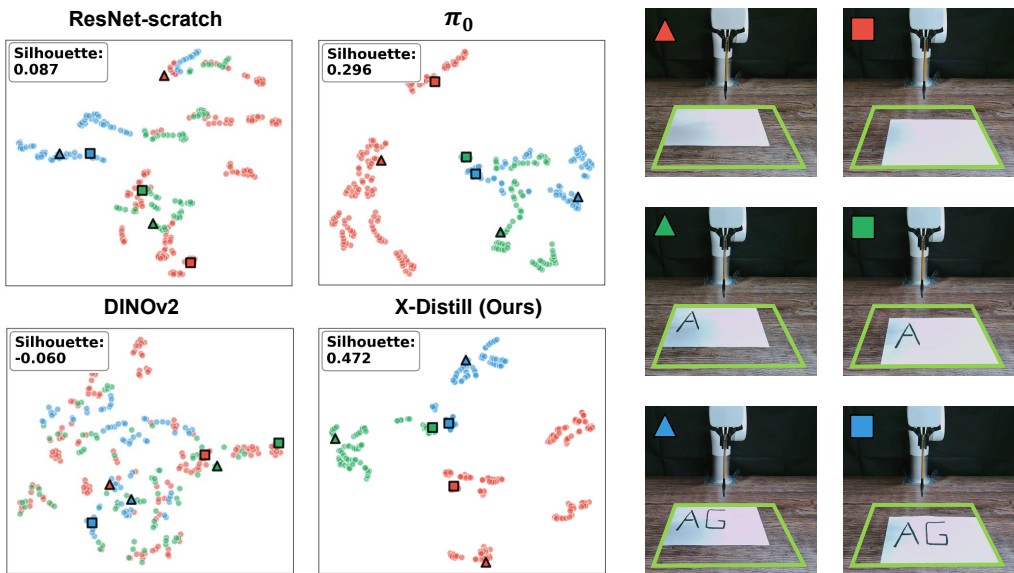

Figure 4: **t-SNE visualization of learned feature spaces on the "Writing AGI" task.** Our X-Distill encoder learns to form three distinct clusters corresponding to the task's semantic stages, quantitatively confirming a well-separated feature space with a high Silhouette Score Rousseeuw (1987) of 0.472, which indicates a high degree of cluster cohesion and separation compared with the baselines. This semantic separability is crucial for the policy to accurately identify the current task stage, enabling precise long-horizon planning for the sequential writing task.

## 5.3 QUALITATIVE ANALYSIS

We focus on our most challenging long-horizon task, **Writing "AGI"**, where success critically depends on the encoder's ability to discern subtle but crucial visual state changes. For instance, before starting to write 'G', the robot's physical state is nearly identical to when it starts writing 'A'; the only distinguishing information is the visual context of the letter 'A' already present on the paper.

Empirically, we observe that Diffusion Policy with ResNet-scratch often fails by repeatedly executing the trajectory for 'A', indicating an inability to visually differentiate these critical semantic states. Meanwhile, Diffusion Policy with DINOv2 and $\pi_0$ (SFT) often get stuck and start trembling before writing any letter, which is a potential sign of underfitting. In comparison, Diffusion Policy with X-Distill is the only one among the compared methods that manages to differentiate all critical stages and completes writing all the three letters sequentially one by one. Even under severe external disturbances that drag the paper away during the writing process, the X-Distill-empowered policy is still robust, responsively following the movement of the paper and rapidly adapting to the correct position for writing the next letter. Example trajectories are shown in Figure 3. To provide deeper insights into how X-Distill achieves superior quantitative performance compared to its counterparts, we conduct further t-SNE analysis and saliency map visualization of the learned visual representations.

**t-SNE visualization of feature space separability.** The global structure of the learned feature space is visualized via t-SNE Maaten & Hinton (2008) in Figure 4. Each data point corresponds to the feature of a frame sampled from three crucial stages for policy decision making marked in three respective colors: (1) before writing 'A', (2) before writing 'G', and (3) before writing 'I'. The features produced from an ideal visual encoder should form three distinct clusters, corresponding to the three colors. It can be observed that X-Distill gives a more separable feature space than the Paligemma Beyer et al. (2024) encoder extracted from $\pi_0$, while the features from both ResNet-scratch and DINOv2 are nearly indistinguishable. These results indicate that X-Distill learns a feature space that is semantically coherent and robust to visual distractors.

**Inspecting task-relevant feature attribution via saliency maps.** To further investigate how our X-Distill achieves emergent semantic feature separation, we inspect pixel-level feature attribution by visualizing the saliency maps. For saliency visualization, we adopt Grad-CAM for CNN-based

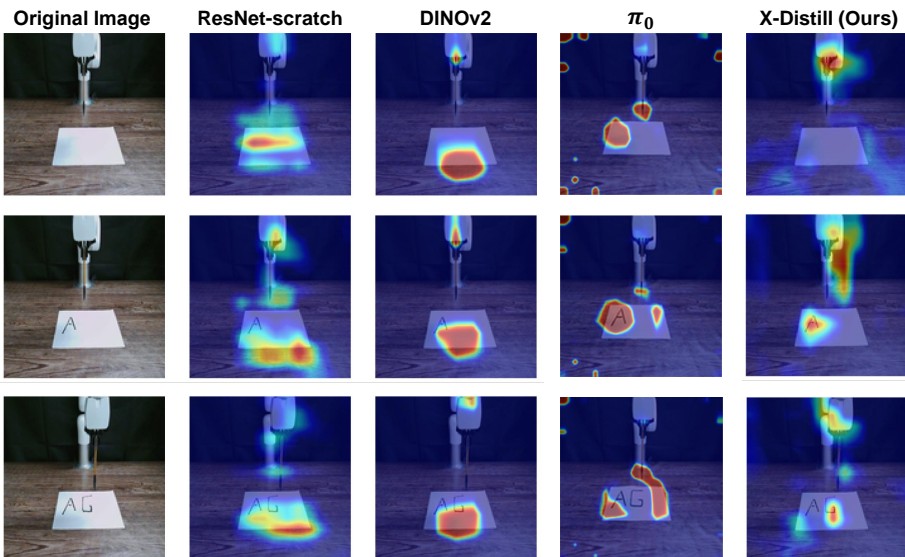

Figure 5: **Saliency map comparison on the "Writing AGI" task.** We visualize the model's visual focus at the beginning of each writing stage. Our X-Distill encoder correctly shifts its attention from the gripper (before 'A'), to the letter 'A' (before 'G'), and finally to the letter 'G' (before 'I'). Baseline models exhibit diffuse or irrelevant attention.

models Selvaraju et al. (2019), and the cross-attention strengths between the `[CLS]` token and all local patch features for ViT-based models Dosovitskiy et al. (2020); Chefer et al. (2021), providing a cross-architecture comparison of the visual focus. As shown in Figure 5, both DINOv2 and $\pi_0$ are unable to effectively shift the high-attention regions throughout the task progress, which cross-verifies our earlier judgment of underfitting. Meanwhile, the saliency maps of ResNet-scratch and X-Distill exhibits more reasonable shifting patterns, but the latter is significantly more precise. More specifically, before writing 'A' on the blank page, the full attention of X-Distill is focused on the **robot gripper**, the primary actor. Then, before writing 'G', its focus dynamically shifts to the **letter 'A'** already on the paper. Finally, before writing 'I', X-Distill shifts attention again, attending to the **letter 'G'**, whose appearance serves as the cue to write the final letter 'I'.

The t-SNE and saliency map visualizations combined reveal that X-Distill successfully learns a semantically meaningful and robust visual representation. Such representation can well differentiate critical states and dynamically focus on task-relevant visual cues, which ultimately contributes to the policy's success in complex long-horizon manipulation tasks.

## 6 CONCLUSION

We introduced **X-Distill**, a framework addressing the trade-off between ViT generalization and CNN sample efficiency. By distilling DINOv2 features into a ResNet-18 on ImageNet, we obtain a robust encoder for data-scarce robotics. Extensive experiments on 34 simulated and 5 real-world tasks show X-Distill outperforms standard baselines and even privileged 3D or VLA policies. Our analysis attributes this to the learned semantically separable feature space. Ultimately, X-Distill demonstrates that a simple, well-founded distillation strategy is a key enabler for data-efficient visuomotor learning.

**Limitations and Future Work.** While effective, our direct feature distillation leaves room for exploration. Future directions include adopting sophisticated techniques to align intermediate features Liu et al. (2022a) and distilling from multimodal VLA teachers to incorporate language priors. Additionally, while we focus on the data-scarce regime, investigating X-Distill's scalability in data-rich scenarios and its application to dynamic tasks like mobile manipulation remain important open questions.

**LLM usage disclosure.** We used the Gemini 2.5 pro model to refine grammar and phrasing.

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

# A  ALGORITHM PSEUDO-CODE

Algorithm 1 outlines our two-stage framework. Sub-algorithm (a) describes the knowledge distillation process where a student ResNet is trained to mimic the representations of a frozen DINOv2 teacher on ImageNet, yielding our X-Distill encoder. Sub-algorithm (b) shows how this pre-trained encoder is fine-tuned with a diffusion policy head on target robotic datasets for effective policy learning.

---

**Algorithm 1** How to acquire and leverage X-Distill.

---

**(a)** Acquiring X-Distill via Knowledge Distillation

**Require:** Teacher encoder $\mathcal{T}$ (frozen DINOv2), Student encoder $\mathcal{S}$ (from-scratch ResNet), Domain-agnostic dataset $\mathcal{D}_{\text{large}}$ (ImageNet).

1: **for** each training epoch **do**
2:     **for** each batch $x \in \mathcal{D}_{\text{large}}$ **do**
3:         $z_T \leftarrow f_{\mathcal{T}}(x)$.
4:         $z_S \leftarrow f_{\mathcal{S}}(x)$.
5:         $L \leftarrow \mathcal{L}_{\text{KD}}(z_S, \text{sg}(z_T))$    ▷ Eq. (1)
6:         Update student encoder $\mathcal{S}$ via $\nabla_{\mathcal{S}} L$.
7:     **end for**
8: **end for**
9: Save the weights of S as S*
10: **return** X-Distilled encoder weights $\mathcal{S}^*$.

**(b)** Leveraging X-Distill via Policy Finetuning

**Require:** X-Distilled encoder weights $\mathcal{S}^*$, Diffusion policy head $\pi_\theta$, Domain-specific dataset $\mathcal{D}_{\text{robotics}}$.

1: Initialize encoder $\mathcal{S}$ with weights from $\mathcal{S}^*$.
2: **for** each training epoch **do**
3:     **for** each batch $(o, a) \in \mathcal{D}_{\text{robotics}}$ **do**
4:         $z_{img} \leftarrow f_{\mathcal{S}}(x)$.
5:         $c \leftarrow \text{concat}(z_{img}, s)$.
6:         Compute $L_{\text{diff}}$.    ▷ Eq. (2)
7:         Update $\mathcal{S}$ and $\pi_\theta$ via $\nabla_{\mathcal{S},\theta} L_{\text{diff}}$.
8:     **end for**
9: **end for**
10: **return** Trained encoder and policy $(\mathcal{S}^{**}, \pi_\theta^*)$.

---

# B  SIMULATION ENVIRONMENTS

**Simulated tasks.** We collect a diverse set of simulated tasks to systematically evaluate imitation learning algorithms, with a particular focus on robotic manipulation in 2D visual settings. Our benchmark draws on three key environments: MetaWorld (Yu et al., 2020), Adroit (Rajeswaran et al., 2017a), and DexArt (Bao et al., 2023), each offering distinct challenges in vision-based motor control.

MetaWorld provides a suite of robotic manipulation tasks designed for multi-task and meta-reinforcement learning, featuring a variety of object interactions in simulated tabletop scenarios, all observable via RGB images. Adroit focuses on dexterous hand manipulation using a high-DoF simulated human hand, with tasks such as object relocation and in-hand rotation, posing significant challenges in policy learning from pixel inputs. DexArt introduces tasks related to articulated object manipulation and artistic activities, such as painting or tool use, requiring precise and fine-grained visuomotor control.

The full list of included tasks is available in Table 5, 6.

# C  TRAINING DETAILS

Our X-Distill model is trained with the hyperparameter configurations summarized in Table 4.

Table 4: Summary of key hyperparameter configurations

| Parameter Description | Parameter Name | Value |
|---|---|---|
| **Diffusion Process** | | |
| Number of diffusion timesteps | num_train_timesteps | 50 |
| Noise schedule | beta_schedule | squaredcos_cap_v2 |
| Prediction target | prediction_type | epsilon |
| **Network Architecture** | | |
| Feature dimension | feature_dim | 64 |
| U-Net decoder channels | down_dims | [256, 512, 1024] |
| Convolution kernel size | kernel_size | 5 |
| Group normalization groups | n_groups | 8 |
| Condition modulation type | condition_type | film |
| **Training Configuration** | | |
| Batch size | batch_size | 64 |
| Number of epochs | num_epochs | 3000 |
| Base learning rate | lr | 0.0001 |
| Optimizer | optimizer | AdamW |
| Weight decay | weight_decay | 0.000001 |
| Gradient accumulation steps | gradient_accumulate_every | 1 |
| EMA decay | use_ema | true |
| **Data Configuration** | | |
| Observation history steps | n_obs_steps | 2 |
| Prediction horizon | horizon | 4 |
| Action steps | n_action_steps | 4 |
| Data loading workers | num_workers | 8 |
| **Inference** | | |
| Number of denoising steps | num_inference_steps | 16 |

## D  ALL SIMULATION RESULTS

Figure 6 presents the training curves across representative tasks from MetaWorld, Adroit, and DexArt, demonstrating the consistent performance advantages of our X-Distill approach throughout the learning process.

Tables 5 and 6 provide comprehensive quantitative comparisons against strong baselines on all benchmark tasks. Our method achieves superior or competitive performance across the majority of tasks, particularly excelling in challenging manipulation scenarios like handle pulling, peg insertion, and complex multi-step operations. The results highlight X-Distill's robustness across varying task difficulties and embodiment domains.

Tables 7 and 8 present an extensive ablation study examining the impact of different teacher-student architecture combinations. Notably, the DINOv2-L to ResNet-18 configuration emerges as the most effective balance between performance and efficiency, while the consistent superiority of distilled representations over from-scratch training underscores the value of our knowledge distillation approach.

Table 5: **Main results on MetaWorld tasks.** 29 tasks are evenly split into 6 rows (5 tasks per row).

| MetaWorld (Easy) | | | | |
|---|---|---|---|---|
| Alg \Task | Lever Pull | Door Close | Drawer Open | Door Lock | Door Unlock |
| ResNet-scratch | $30 \pm 18$ | $\mathbf{100 \pm 0}$ | $77 \pm 6$ | $70 \pm 19$ | $82 \pm 5$ |
| Theia | $48 \pm 14$ | $\mathbf{100 \pm 0}$ | $\mathbf{100 \pm 0}$ | $47 \pm 3$ | $83 \pm 8$ |
| Depth-Anything | $17 \pm 2$ | $\mathbf{100 \pm 0}$ | $63 \pm 5$ | $48 \pm 2$ | $78 \pm 13$ |
| DINOv2 | $47 \pm 5$ | $\mathbf{100 \pm 0}$ | $\mathbf{100 \pm 0}$ | $63 \pm 2$ | $90 \pm 4$ |
| **X-Distill (Ours)** | $\mathbf{75 \pm 8}$ | $\mathbf{100 \pm 0}$ | $\mathbf{100 \pm 0}$ | $\mathbf{100 \pm 0}$ | $\mathbf{100 \pm 0}$ |
| PointNet-DP3 | $79 \pm 8$ | $100 \pm 0$ | $100 \pm 0$ | $100 \pm 0$ | $100 \pm 0$ |

| MetaWorld (Easy) | | | | |
|---|---|---|---|---|
| Alg \Task | Drawer Close | Faucet Close | Faucet Open | Handle Press | Handle Pull |
| ResNet-scratch | $\mathbf{100 \pm 0}$ | $\mathbf{100 \pm 0}$ | $\mathbf{100 \pm 0}$ | $83 \pm 13$ | $25 \pm 22$ |
| Theia | $\mathbf{100 \pm 0}$ | $3 \pm 3$ | $7 \pm 3$ | $\mathbf{100 \pm 0}$ | $13 \pm 10$ |
| Depth-Anything | $\mathbf{100 \pm 0}$ | $88 \pm 6$ | $\mathbf{100 \pm 0}$ | $85 \pm 7$ | $18 \pm 2$ |
| DINOv2 | $\mathbf{100 \pm 0}$ | $93 \pm 2$ | $\mathbf{100 \pm 0}$ | $\mathbf{100 \pm 0}$ | $28 \pm 5$ |
| **X-Distill (Ours)** | $\mathbf{100 \pm 0}$ | $\mathbf{100 \pm 0}$ | $\mathbf{100 \pm 0}$ | $\mathbf{100 \pm 0}$ | $\mathbf{95 \pm 4}$ |
| PointNet-DP3 | $100 \pm 0$ | $100 \pm 0$ | $100 \pm 0$ | $100 \pm 0$ | $45 \pm 8$ |

| MetaWorld (Easy) | | | | |
|---|---|---|---|---|
| Alg \Task | Handle Pull Side | Plate Slide | Plate Slide Back | Plate Slide Back Side | Plate Slide Side |
| ResNet-scratch | $3 \pm 5$ | $90 \pm 14$ | $\mathbf{100 \pm 0}$ | $\mathbf{100 \pm 0}$ | $\mathbf{100 \pm 0}$ |
| Theia | $12 \pm 16$ | $40 \pm 30$ | $38 \pm 8$ | $62 \pm 47$ | $2 \pm 3$ |
| Depth-Anything | $12 \pm 2$ | $80 \pm 12$ | $\mathbf{100 \pm 0}$ | $\mathbf{100 \pm 0}$ | $\mathbf{100 \pm 0}$ |
| DINOv2 | $48 \pm 5$ | $80 \pm 4$ | $\mathbf{100 \pm 0}$ | $\mathbf{100 \pm 0}$ | $\mathbf{100 \pm 0}$ |
| **X-Distill (Ours)** | $\mathbf{95 \pm 7}$ | $\mathbf{100 \pm 0}$ | $\mathbf{100 \pm 0}$ | $\mathbf{100 \pm 0}$ | $\mathbf{100 \pm 0}$ |
| PointNet-DP3 | $100 \pm 0$ | $100 \pm 0$ | $100 \pm 0$ | $100 \pm 0$ | $100 \pm 0$ |

| MetaWorld (Easy) | | | | |
|---|---|---|---|---|
| Alg \Task | Reach Wall | Window Close | Window Open | Reach | Peg unplug side |
| ResNet-scratch | $\mathbf{77 \pm 2}$ | $\mathbf{100 \pm 0}$ | $93 \pm 10$ | $47 \pm 13$ | $55 \pm 8$ |
| Theia | $67 \pm 3$ | $42 \pm 6$ | $95 \pm 5$ | $48 \pm 6$ | $10 \pm 0$ |
| Depth-Anything | $48 \pm 10$ | $90 \pm 14$ | $70 \pm 4$ | $45 \pm 4$ | $22 \pm 6$ |
| DINOv2 | $53 \pm 6$ | $\mathbf{100 \pm 0}$ | $78 \pm 13$ | $\mathbf{52 \pm 2}$ | $38 \pm 2$ |
| **X-Distill (Ours)** | $73 \pm 6$ | $\mathbf{100 \pm 0}$ | $\mathbf{100 \pm 0}$ | $\mathbf{52 \pm 6}$ | $\mathbf{87 \pm 2}$ |
| PointNet-DP3 | $68 \pm 3$ | $100 \pm 0$ | $100 \pm 0$ | $24 \pm 1$ | $92 \pm 2$ |

| MetaWorld (Medium) | | | | |
|---|---|---|---|---|
| Alg \Task | Coffee Push | Bin picking | Coffee Pull | Push Wall | Peg Insert Side |
| ResNet-scratch | $82 \pm 10$ | $68 \pm 9$ | $55 \pm 4$ | $48 \pm 10$ | $28 \pm 13$ |
| Theia | $32 \pm 3$ | $10 \pm 0$ | $2 \pm 3$ | $3 \pm 6$ | $0 \pm 0$ |
| Depth-Anything | $38 \pm 2$ | $32 \pm 2$ | $47 \pm 6$ | $33 \pm 2$ | $13 \pm 2$ |
| DINOv2 | $35 \pm 0$ | $52 \pm 13$ | $52 \pm 6$ | $48 \pm 6$ | $35 \pm 4$ |
| **X-Distill (Ours)** | $\mathbf{97 \pm 5}$ | $\mathbf{95 \pm 4}$ | $\mathbf{95 \pm 4}$ | $\mathbf{80 \pm 0}$ | $\mathbf{88 \pm 2}$ |
| PointNet-DP3 | $95 \pm 4$ | $65 \pm 19$ | $85 \pm 11$ | $49 \pm 8$ | $72 \pm 5$ |

| MetaWorld (Medium / Hard / Very Hard) | | | |
|---|---|---|---|
| Alg \Task | Sweep | Sweep into | Pick out of hole | Disassemble |
| ResNet-scratch | $22 \pm 2$ | $33 \pm 9$ | $38 \pm 9$ | $50 \pm 29$ |
| Theia | $12 \pm 3$ | $37 \pm 3$ | $0 \pm 0$ | $38 \pm 6$ |
| Depth-Anything | $20 \pm 4$ | $22 \pm 6$ | $42 \pm 10$ | $43 \pm 6$ |
| DINOv2 | $48 \pm 5$ | $52 \pm 5$ | $\mathbf{48 \pm 9}$ | $38 \pm 5$ |
| **X-Distill (Ours)** | $\mathbf{85 \pm 4}$ | $\mathbf{78 \pm 5}$ | $\mathbf{48 \pm 6}$ | $\mathbf{88 \pm 6}$ |
| PointNet-DP3 | $83 \pm 5$ | $45 \pm 16$ | $14 \pm 9$ | $72 \pm 6$ |

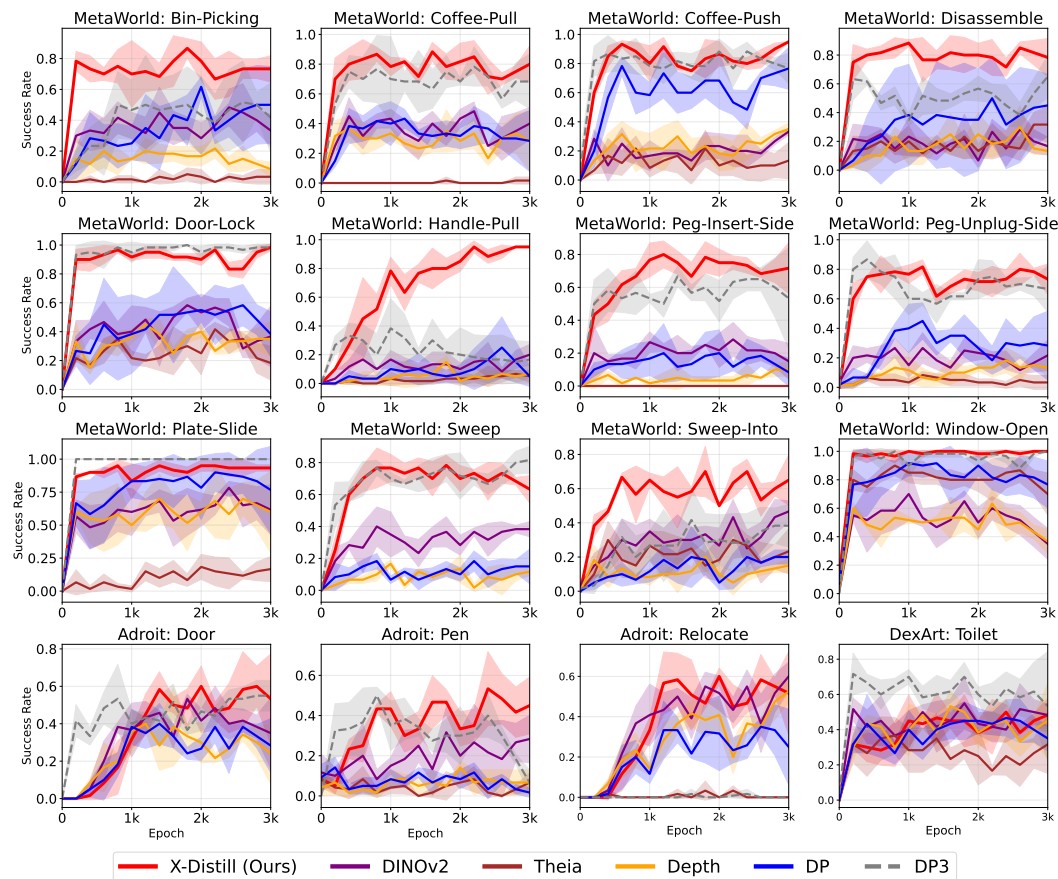

Figure 6: **Training curves on representative simulation tasks.** Success rates are shown for selected tasks from MetaWorld, Adroit, and DexArt.

Table 6: **Main results on Adroit and Dexart tasks.** Tasks are grouped by difficulty and arranged across five visually aligned rows.

| Alg \ Task | Adroit | | | Dexart | |
|---|---|---|---|---|---|
| | Door | Pen | Relocate | Laptop | Toilet |
| ResNet-scratch | $47 \pm 7$ | $18 \pm 2$ | $48 \pm 8$ | $52 \pm 5$ | $57 \pm 2$ |
| Theia | $7 \pm 2$ | $14 \pm 1$ | $5 \pm 4$ | $0 \pm 0$ | $48 \pm 2$ |
| Depth-Anything | $52 \pm 6$ | $16 \pm 1$ | $53 \pm 5$ | $70 \pm 4$ | $62 \pm 2$ |
| DINOv2 | $57 \pm 6$ | $38 \pm 11$ | $60 \pm 7$ | $53 \pm 13$ | $\mathbf{63 \pm 5}$ |
| **X-Distill (Ours)** | $\mathbf{73 \pm 9}$ | $\mathbf{60 \pm 11}$ | $\mathbf{72 \pm 5}$ | $\mathbf{65 \pm 4}$ | $62 \pm 2$ |
| PointNet-DP3 | $67 \pm 6$ | $52 \pm 6$ | $3 \pm 2$ | $90 \pm 4$ | $80 \pm 0$ |

Table 7: **Main results on MetaWorld tasks.** Tasks are grouped by difficulty and arranged across visually aligned sections.

| | | **Metaworld - Easy Tasks** | | | |
|---|---|---|---|---|---|
| Teacher | Student | Lever Pull | Door Close | Drawer Open | Door Lock |
| DINOv2-L | ResNet-18 (11M) | $75 \pm 8$ | $\mathbf{100 \pm 0}$ | $\mathbf{100 \pm 0}$ | $\mathbf{100 \pm 0}$ |
| | ViT-S-Half (11M) | $40 \pm 4$ | $\mathbf{100 \pm 0}$ | $\mathbf{100 \pm 0}$ | $68 \pm 10$ |
| | ConvNeXt (89M) | $73 \pm 8$ | $\mathbf{100 \pm 0}$ | $\mathbf{100 \pm 0}$ | $\mathbf{100 \pm 0}$ |
| DINOv2-S | ResNet-18 (11M) | $\mathbf{82 \pm 12}$ | $\mathbf{100 \pm 0}$ | $\mathbf{100 \pm 0}$ | $\mathbf{100 \pm 0}$ |

| | | **Metaworld - Easy Tasks** | | | |
|---|---|---|---|---|---|
| Teacher | Student | Door Unlock | Drawer Close | Faucet Close | Faucet Open |
| DINOv2-L | ResNet-18 (11M) | $\mathbf{100 \pm 0}$ | $\mathbf{100 \pm 0}$ | $\mathbf{100 \pm 0}$ | $\mathbf{100 \pm 0}$ |
| | ViT-S-Half (11M) | $68 \pm 2$ | $\mathbf{100 \pm 0}$ | $22 \pm 17$ | $\mathbf{100 \pm 0}$ |
| | ConvNeXt (89M) | $\mathbf{100 \pm 0}$ | $\mathbf{100 \pm 0}$ | $\mathbf{100 \pm 0}$ | $\mathbf{100 \pm 0}$ |
| DINOv2-S | ResNet-18 (11M) | $\mathbf{100 \pm 0}$ | $\mathbf{100 \pm 0}$ | $\mathbf{100 \pm 0}$ | $\mathbf{100 \pm 0}$ |

| | | **Metaworld - Easy Tasks** | | | |
|---|---|---|---|---|---|
| Teacher | Student | Handle Press | Handle Pull | Handle Pull Side | Plate Slide |
| DINOv2-L | ResNet-18 (11M) | $\mathbf{100 \pm 0}$ | $95 \pm 4$ | $\mathbf{95 \pm 7}$ | $\mathbf{100 \pm 0}$ |
| | ViT-S-Half (11M) | $98 \pm 2$ | $13 \pm 5$ | $10 \pm 0$ | $93 \pm 2$ |
| | ConvNeXt (89M) | $\mathbf{100 \pm 0}$ | $80 \pm 15$ | $78 \pm 6$ | $98 \pm 2$ |
| DINOv2-S | ResNet-18 (11M) | $\mathbf{100 \pm 0}$ | $\mathbf{98 \pm 3}$ | $90 \pm 13$ | $\mathbf{100 \pm 0}$ |

| | | **Metaworld - Easy Tasks** | | | |
|---|---|---|---|---|---|
| Teacher | Student | Plate Slide Back | Plate Slide Back Side | Plate Slide Side | Reach Wall |
| DINOv2-L | ResNet-18 (11M) | $\mathbf{100 \pm 0}$ | $\mathbf{100 \pm 0}$ | $\mathbf{100 \pm 0}$ | $73 \pm 6$ |
| | ViT-S-Half (11M) | $85 \pm 7$ | $\mathbf{100 \pm 0}$ | $\mathbf{100 \pm 0}$ | $72 \pm 5$ |
| | ConvNeXt (89M) | $\mathbf{100 \pm 0}$ | $\mathbf{100 \pm 0}$ | $\mathbf{100 \pm 0}$ | $67 \pm 3$ |
| DINOv2-S | ResNet-18 (11M) | $\mathbf{100 \pm 0}$ | $\mathbf{100 \pm 0}$ | $\mathbf{100 \pm 0}$ | $\mathbf{75 \pm 5}$ |

| | | **Metaworld - Easy Tasks** | | | |
|---|---|---|---|---|---|
| Teacher | Student | Window Close | Window Open | Coffee Push | Bin Picking |
| DINOv2-L | ResNet-18 (11M) | $\mathbf{100 \pm 0}$ | $\mathbf{100 \pm 0}$ | $\mathbf{97 \pm 5}$ | $\mathbf{95 \pm 4}$ |
| | ViT-S-Half (11M) | $92 \pm 8$ | $95 \pm 4$ | $35 \pm 4$ | $10 \pm 4$ |
| | ConvNeXt (89M) | $\mathbf{100 \pm 0}$ | $\mathbf{100 \pm 0}$ | $88 \pm 8$ | $88 \pm 6$ |
| DINOv2-S | ResNet-18 (11M) | $\mathbf{100 \pm 0}$ | $\mathbf{100 \pm 0}$ | $95 \pm 5$ | $85 \pm 9$ |

| | | **Metaworld - Medium Tasks** | | | |
|---|---|---|---|---|---|
| Teacher | Student | Reach | Peg Unplug Side | Coffee Pull | Push Wall |
| DINOv2-L | ResNet-18 (11M) | $\mathbf{52 \pm 6}$ | $87 \pm 2$ | $95 \pm 4$ | $80 \pm 0$ |
| | ViT-S-Half (11M) | $50 \pm 4$ | $33 \pm 2$ | $23 \pm 18$ | $37 \pm 6$ |
| | ConvNeXt (89M) | $52 \pm 3$ | $\mathbf{88 \pm 8}$ | $83 \pm 3$ | $73 \pm 8$ |
| DINOv2-S | ResNet-18 (11M) | $\mathbf{52 \pm 6}$ | $\mathbf{88 \pm 6}$ | $\mathbf{95 \pm 0}$ | $\mathbf{83 \pm 3}$ |

Table 8: **Main results on MetaWorld tasks (continued).**

| Metaworld - Medium/Hard Tasks | | | | | |
|---|---|---|---|---|---|
| Teacher | Student | Peg Insert Side | Sweep | Sweep Into | Pick Out of Hole |
| DINOv2-L | ResNet-18 (11M) | $88 \pm 2$ | $85 \pm 4$ | $78 \pm 5$ | $48 \pm 6$ |
| | ViT-S-Half (11M) | $7 \pm 6$ | $45 \pm 8$ | $20 \pm 4$ | $2 \pm 2$ |
| | ConvNeXt (89M) | $57 \pm 16$ | $75 \pm 5$ | $\mathbf{78 \pm 6}$ | $\mathbf{50 \pm 5}$ |
| DINOv2-S | ResNet-18 (11M) | $\mathbf{90 \pm 5}$ | $\mathbf{85 \pm 5}$ | $78 \pm 3$ | $43 \pm 16$ |

| Very Hard Tasks | | |
|---|---|---|
| Teacher | Student | Disassemble |
| DINOv2-L | ResNet-18 (11M) | $88 \pm 6$ |
| | ViT-S-Half (11M) | $40 \pm 7$ |
| | ConvNeXt (89M) | $83 \pm 8$ |
| DINOv2-S | ResNet-18 (11M) | $\mathbf{90 \pm 5}$ |

## E    REAL-WORLD SETUP

Our real-world evaluation setup, depicted in Figure 7, features a 6-DoF X-Arm robot manipulator controlled via policies trained with our method. The system utilizes a UGreen camera for visual perception and a Meta-Quest headset for human demonstration data collection, enabling comprehensive evaluation of manipulation capabilities in physical environments.

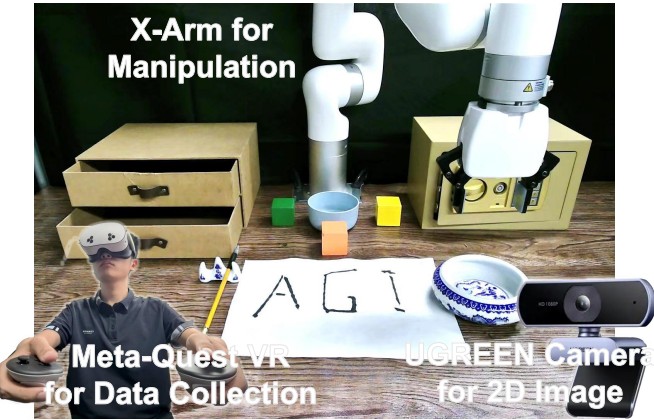

Figure 7: **Hardware setup** comprising the 6-DoF X-Arm robot for control, Meta-Quest headset for data collection, and UGreen camera for visual perception.

## F    KEYFRAME SEQUENCE FROM REAL-WORLD TASK

The following figures present keyframe sequences from our real-world robotic manipulation experiments, illustrating both successful executions and representative failure cases for each task.

**Move Cube Task:** Successful completion requires the robot to reliably pick up the cube and place it into the bowl. Common failure modes include the gripper colliding with the cube, triggering emergency stops due to excessive force, or failing to establish a secure grasp on the object (see Figure 8).

**Writing "AGI" Task:** Success is defined by the accurate writing of the letters "AGI". Typical failures include writing incorrect characters, repeatedly writing the same letter without progression, or complete failure to produce any legible writing (see Figure 9).

**Move Cube**

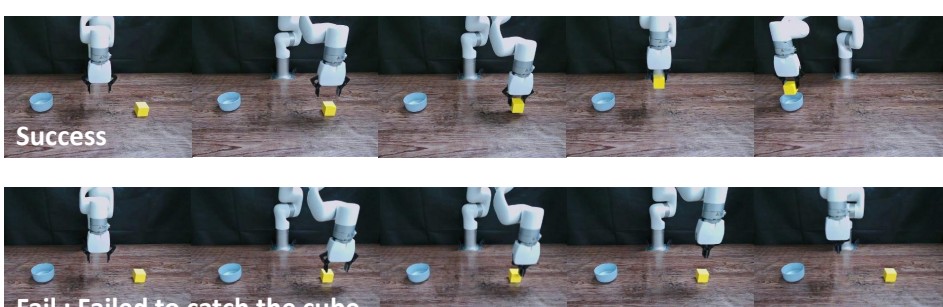

Figure 8: **Move Cube task keyframe sequence.** Illustrates successful executions and typical failure cases of the Move Cube task.

**Writing "AGI"**

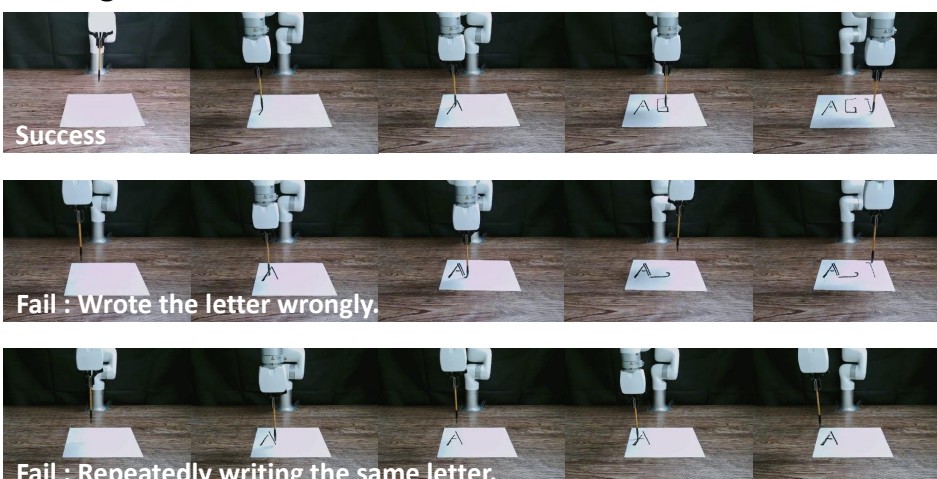

Figure 9: **Writing "AGI" task keyframe sequence.** Illustrates successful executions and typical failure cases of the writing "AGI" task.

**Move Brush Task:** Successful execution involves correctly orienting and placing the brush into the holder. Common failures include incorrect brush orientation, collisions with the brush handle triggering emergency stops, or complete misses when attempting to grasp the brush (see Figure 10).

**Move Brush**

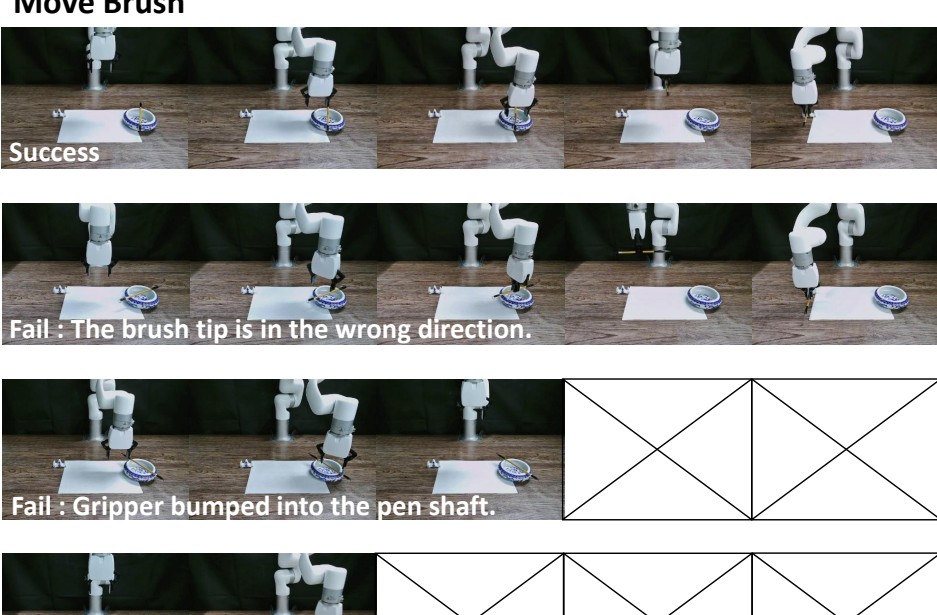

Figure 10: **Move Brush task keyframe sequence.** Illustrates successful executions and typical failure cases of the move brush task.

**Drawer Open Task:** Success requires smoothly opening the drawer without collisions. The primary failure mode involves the gripper colliding with the drawer edges, preventing proper engagement and manipulation (see Figure 11).

**Drawer Open**

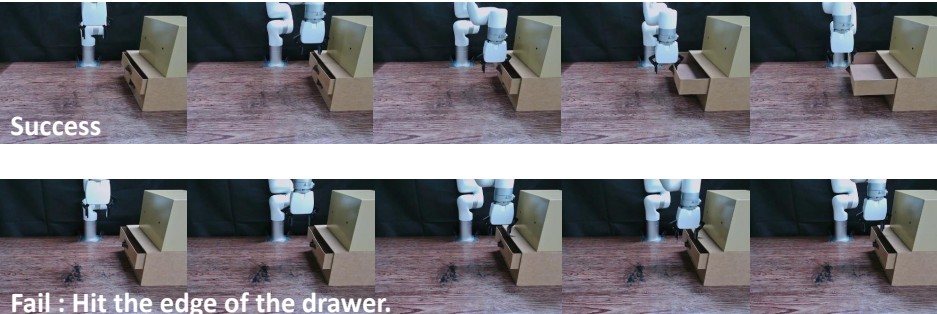

Figure 11: **Drawer Open task keyframe sequence.** Illustrates successful executions and typical failure cases of the drawer open task.

**Door Close Task:** Successful completion entails securely closing the door. The main failure occurs when the gripper fails to make proper contact with the door surface, resulting in inability to initiate or complete the closing motion (see Figure 12).

**Door Close**

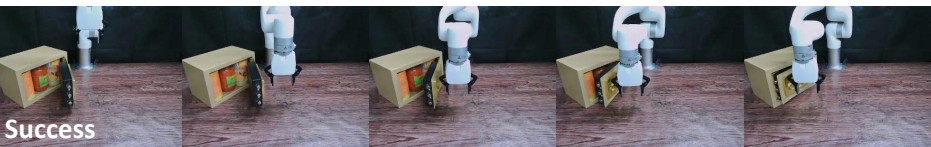

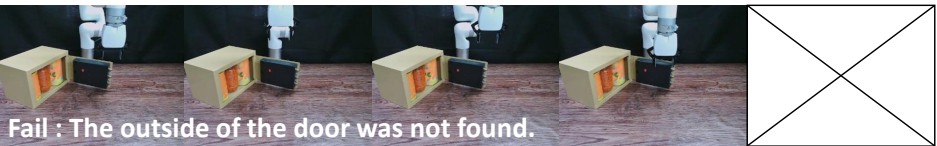

Figure 12: **Door Close task keyframe sequence.** Illustrates successful executions and typical failure cases of the door close task.

## G   ADDITIONAL EXPERIMENTAL RESULTS

In this section, we provide comprehensive results from additional experiments conducted to investigate the impact of different pre-training objectives, parameter-efficient fine-tuning methods, and data scaling laws.

### G.1   COMPARISON WITH R3M AND PEFT (LoRA)

To further validate the effectiveness of our X-Distill framework, we compared it against two important baselines on the 29 MetaWorld tasks (10 demonstrations per task):

- **R3M** (Nair et al., 2022): A ResNet-18 encoder pre-trained on Ego4D human videos. This comparison isolates the effect of the pre-training objective (video-language vs. our distillation) since the architecture is identical to ours.
- **DINOv2 + LoRA**: We applied Low-Rank Adaptation (LoRA) (Hu et al., 2022) to the DINOv2-Small encoder (rank=16, applied to q, k, v matrices) during policy learning. This tests whether parameter-efficient fine-tuning can bridge the gap between ViTs and CNNs in the low-data regime.

The results are summarized in Table 9.

Table 9: Comparison against R3M (alternative pre-training) and LoRA (PEFT) on MetaWorld tasks (10 demos). X-Distill significantly outperforms both, highlighting the superiority of our distillation objective and architectural choice.

| Method | Configuration | Success Rate (%) |
|---|---|---|
| **X-Distill (Ours)** | ResNet-18 (Distilled) | **90.8** |
| DINOv2 (Full Finetune) | ViT-S (Pre-trained) | 68.2 |
| **DINOv2 + LoRA** | ViT-S (PEFT, rank=16) | 74.5 |
| ResNet-scratch | ResNet-18 (Random Init) | 67.4 |
| **R3M** | ResNet-18 (Frozen, Ego4D) | 14.2 |

**Analysis.** First, **R3M** achieves a low success rate of 14.2%, significantly underperforming even the scratch-trained ResNet. This suggests that while R3M's representations are temporally smooth for video analysis, they lack the fine-grained spatial precision required for manipulation control, validating that our specific distillation from DINOv2 is critical. Second, while **LoRA** (74.5%) indeed improves over full fine-tuning (68.2%) by mitigating catastrophic forgetting, it still significantly underperforms X-Distill (90.8%). This refutes the hypothesis that PEFT is the optimal solution for this specific low-data regime and confirms that the architectural inductive bias of the CNN student is the dominant factor for success.

## G.2  DATA SCALING ANALYSIS

To understand the scalability of our method and identify the potential "crossover point" where ViTs might outperform CNNs, we conducted a data scaling experiment on the real-world "Move Brush" task. We trained policies using $N = \{20, 40, 60, 80, 100\}$ demonstrations. The results are visualized in Figure 13 and detailed in Table 10.

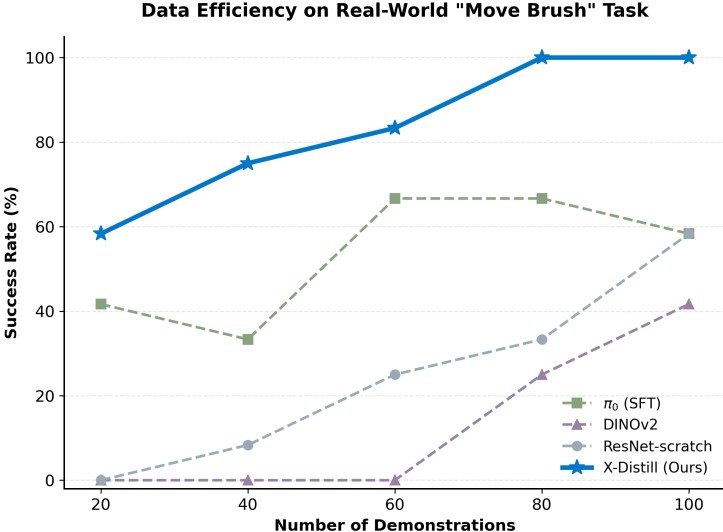

Figure 13: **Data scaling efficiency analysis.** X-Distill (Blue) demonstrates superior data efficiency, saturating at 100% success with only 80 demos. ViT-based baselines start slowly, and generalist VLAs ($\pi_0$) show instability.

Table 10: Detailed success rates (out of 12 trials) and percentages for the Data Scaling experiment on the "Move Brush" task.

| Method | Number of Demonstrations | | | | |
|---|---|---|---|---|---|
| | 20 | 40 | 60 | 80 | 100 |
| ResNet-scratch | 0.0% | 8.3% | 25.0% | 33.3% | 58.3% |
| DINOv2 | 0.0% | 0.0% | 0.0% | 25.0% | 41.7% |
| $\pi_0$ (SFT) | 41.7% | 33.3% | 66.7% | 66.7% | 58.3% |
| **X-Distill (Ours)** | **58.3%** | **75.0%** | **83.3%** | **100.0%** | **100.0%** |

**Analysis.** X-Distill demonstrates extreme data efficiency, yielding workable policies ($> 50\%$) with as few as 20 demos and reaching perfect performance at 80 demos. In contrast, the ViT baseline (DINOv2) requires significantly more data to begin learning, only reaching 41.7% at 100 demos. This trend suggests that the crossover point where a ViT might outperform our distilled CNN lies at thousands of trajectories, far beyond the typical data regime for accessible robotic learning.

