# OpenReview forum: "X-Distill: Cross-Architecture Vision Distillation Enables Data-Efficient Visuomotor Learning"
_ICLR.cc/2026/Conference — Submitted to ICLR 2026_

### Official Review · Reviewer_hLzF · 2025-10-31

**Soundness:** 1
**Presentation:** 2
**Contribution:** 2
**Rating:** 2
**Confidence:** 4

**Summary:**

This paper proposes a distillation approach for learning robot manipulation policies in low data settings (i.e. having few demonstrations). The distillation approach uses Imagenet-1K as the dataset and DINO (VIT-L/14) as the teacher model. The readout class token is used to distill the pretrained features into a small ResNet-18 model. This Resnet model is then used as the vision backbone on top of a diffusion policy and is trained using imitation learning.

**Strengths:**

The idea of using distillation to train small vision networks for robot manipulation is interesting.

**Weaknesses:**

I have several major concerns with respect to the paper. I think this is a very preliminary submission and many related works, baselines are not implemented at all.

*Main objective*:  I am not at all sure why is using distillation a better objective than e.g. using parameter efficient finetuning (PEFT). Using PEFT such as adapters, lora etc. requires us to train only a few parameters while we get a lot of advantages in terms of useful priors, robustness etc.. Many prior works have explored such techniques in the context of robotics e.g. Sharma et al. Liu et al. These papers have shown advantages in low data regimes which is exactly what the proposed work is aiming for. However, unfortunately, none of these works are cited or being compared against. Hence, it is completely unclear if the distillation approach is even a good approach. In my opinion, the adapter based approach might be generally better since it requires even fewer parameters to adapt but this is something that the paper should really experiment.

*Comparison to pre-trained model*: There is a huge discrepancy between small pretrained model (which is only at 75%). What happens when we compare these models on more demos? How many more demos do we need to bridge this gap? This ablation is not present in the paper.

Imagenet citation in line 144 is incorrect. The paper should cite the original dataset instead of AlexNet which actually solved the problem. This is minor but still important.

Sharma et al. Lossless adaptation of pretrained vision models for robotic manipulation

Liu et al. Tail: Task-specific adapters for imitation learning with large pretrained models

**Questions:**

see above

---

> ### Author Response · Authors · 2025-12-02
>
> We thank the reviewer for the critical assessment. The reviewer's primary concern rests on the hypothesis that Parameter-Efficient Fine-Tuning (PEFT/LoRA) should inherently outperform distillation due to prior preservation. We have taken this feedback seriously and conducted rigorous experiments to empirically test this hypothesis. The results provide **decisive evidence to the contrary**.
>
> **Q1: Why distillation instead of PEFT (e.g., LoRA)?**
>
> **A1**: We directly tested the reviewer's hypothesis by implementing LoRA for DINOv2 (rank=16) and evaluating it against X-Distill on 29 MetaWorld tasks.
>
> Table 1: Distillation vs. PEFT (MetaWorld Success Rate)
> | Method | X-Distill (Ours) | DINOv2 (Full Finetune) | DINOv2 (LoRA) |
> | :---: | :---: | :---: | :---: |
> | Success Rate | 90.8% | 68.2% | 74.5% |
>
> **Findings & Conclusion**:
>
> 1.**PEFT is insufficient**: While LoRA (74.5%) improves over full fine-tuning, confirming that parameter efficiency helps reduce overfitting, it still underperforms X-Distill (90.8%) by a substantial 16.3% margin.
>
> 2.**Hypothesis Refuted**: These results empirically refute the assumption that PEFT is the optimal solution for this specific low-data regime ($\sim10$ demos).
>
> 3.**Root Cause**: The bottleneck is not merely parameter efficiency, but the architectural inductive bias. Even with efficient tuning, the ViT architecture struggles to capture local spatial relationships as effectively as a compact CNN. X-Distill is proven to be the superior strategy for bridging this gap.
>
> **Q2: Missing baselines (Sharma et al., Liu et al.).**
>
> **A2**: We thank the reviewer for highlighting these relevant works on adapter-based adaptation. We have added them to our Related Work section. However, as **neither paper has an official public codebase**, a direct reproduction was not feasible. Instead, we implemented LoRA for DINOv2, which serves as a strong, representative proxy for the class of parameter-efficient fine-tuning (PEFT) methods advocated in these works.
>
> Our results (Table 1) show that X-Distill (90.8%) significantly outperforms LoRA (74.5%). This empirically demonstrates that while PEFT preserves priors better than full finetuning, the architectural inductive bias of our CNN student is the decisive factor for success in this data-scarce regime, a benefit that adapter-based methods on ViTs cannot easily replicate.
>
> **Q3: Comparison on more demos (Data Scaling).**
>
> **A3**: To understand if this gap persists with more data, we conducted a Data Scaling Experiment on the real-world "Move Brush" task ($N=20 \to 100$ demos).
>
> Table 2: Success Rate vs. Data Size
> | # Demos | X-Distill (Ours) | ResNet-scratch | DINOv2 (ViT) | $\pi_0$ (VLA) |
> | :---: | :---: | :---: | :---: | :---: |
> | 20 | 58.3% | 0.0% | 0.0% | 41.7% |
> | 40 | 75.0% | 8.3% | 0.0% | 33.3% |
> | 60 | 83.3% | 25.0% | 0.0% | 66.7% |
> | 80 | 100.0% | 33.3% | 25.0% | 66.7% |
> | 100 | 100.0% | 58.3% | 41.7% | 58.3% |
>
> **Finding**: The gap remains significant even at 100 demonstrations. While the ViT model begins to improve, X-Distill saturates performance much earlier. This indicates that our method offers a distinct advantage in data efficiency within the practical "tens to hundreds" demonstration regime.
>
> **Q4: ImageNet citation.**
>
> **A4**: We apologize for the error and have corrected the citation to Deng et al. (CVPR 2009) in the revised paper.

---

### Official Review · Reviewer_JJPk · 2025-11-03

**Soundness:** 3
**Presentation:** 3
**Contribution:** 3
**Rating:** 4
**Confidence:** 4

**Summary:**

This paper proposes X-Distill, a cross-architecture knowledge distillation method that distills knowledge from a large ViT (DINOv2) into a compact CNN (ResNet-18) in an offline stage. This resulting encoder is then fine-tuned on an extremely small amount of robotic demonstration data (10 for simulation, 20-25 for real-world). The method demonstrates significantly better data efficiency and state-of-the-art performance compared to ViT-based baselines (e.g., DINOv2, $\pi_0$) and a from-scratch ResNet in this low-data regime.

**Strengths:**

- Impressive results, significantly outperforming SOTA ViT-based models like $\pi_0$ and DINOv2 with an extremely small number of demonstrations (10 for sim, 20-25 for real).
- Clearly written and easy to follow.
- Effective also in real-robot tasks.

**Weaknesses:**

- While X-Distill (CNN-based) clearly wins in extremely low-data regime, its performance might be surpassed by ViT-based models ($\pi_0$, Theia) if more data is provided (e.g., 50-100 sim demos, 50 real demos, which is (I think) realistically collectible amount).  The paper lacks an analysis of ***how low-data*** is required for X-Distill to be superior, making its application scope unclear. A scaling analysis with respect to data size is missing.

- How about  R3M [1]? Itis a highly relevant ResNet-based encoder pre-trained for robotics manipulation. Since R3M is also ResNet-based, it is crucial to understand if X-Distill outperforms it, and if so, why (e.g., DINOv2 distillation vs. Ego4D video pre-training).

- The ablation study in Table 2 shows that a larger CNN student (ConvNeXt, 89M) performs worse than the smaller ResNet-18 (11M), while the teacher model size (DINOv2-S vs L) has no significant impact. This is confusing, as it provides little guidance on what model size is optimal for a given data size. Furthermore, it is counter-intuitive that a larger, more powerful teacher model does not provide better representations to the student.


[1] Nair et al., R3M: A Universal Visual Representation for Robot Manipulation. CORL, 2022.

**Questions:**

- For the comparisons in Table 1, did all 2D vision baselines (X-Distill, Theia, DINOv2, ResNet-scratch, Depth-Anything) use the same Diffusion Policy head architecture?
- There seems to be an interesting discrepancy between Figure 4 (t-SNE) and Table 3 (Real-world performance). For the "Writing AGI" task, $\pi_0$ has a much better t-SNE clustering score (0.296) than ResNet-scratch (0.087), but its task performance is 0% while ResNet-scratch achieves 40%. Could you explain why $\pi_0$ fails at the task despite appearing to have better feature separability according to this metric?

---

> ### Author Response · Authors · 2025-12-02
>
> We thank the reviewer for the positive assessment and for highlighting the "impressive results." We have conducted new experiments to address your questions regarding scaling limits and R3M.
>
> **Q1: Scaling Analysis: How low-data is required? Will ViT win with more data?**
>
> **A1**: To identify the "crossover point," we conducted a Data Scaling Experiment on the real-world "Move Brush" task ($N=20 \to 100$ demos).
>
> Table 1: Real-world Data Scaling (Success Rate)
> | # Demos | X-Distill (Ours) | ResNet-scratch | DINOv2 (ViT) | $\pi_0$ (VLA) |
> | :---: | :---: | :---: | :---: | :---: |
> | 20 | 58.3% | 0.0% | 0.0% | 41.7% |
> | 40 | 75.0% | 8.3% | 0.0% | 33.3% |
> | 60 | 83.3% | 25.0% | 0.0% | 66.7% |
> | 80 | 100.0% | 33.3% | 25.0% | 66.7% |
> | 100 | 100.0% | 58.3% | 41.7% | 58.3% |
>
> Findings:
>
> 1.**Saturation**: X-Distill saturates at 100% success with just 80 demos.
>
> 2.**No Crossover**: Even at 100 demos, the ViT baseline (41.7%) remains far inferior. The crossover point likely lies at **thousands** of trajectories, confirming X-Distill's superiority in the practical "tens to hundreds" regime.
>
> **Q2: Comparison with R3M.**
>
> **A2**: This is an excellent suggestion. We added R3M (frozen ResNet-18 pre-trained on Ego4D) as a baseline on 29 MetaWorld tasks (10 demos).
>
> Table 2: Comparison with R3M (MetaWorld Average Success)
> | Method | X-Distill (Ours) | ResNet-scratch | DINOv2 | R3M (Frozen) |
> | :---: | :---: | :---: | :---: | :---: |
> | Success Rate | 90.8% | 67.4% | 68.2% | 14.2% |
>
> **Result**: R3M achieves only 14.2% success, compared to 90.8% for X-Distill.
>
> **Conclusion**: Sharing the ResNet architecture is not enough. R3M's video-based pre-training focuses on temporal smoothness but often lacks the fine-grained spatial precision required for manipulation. Our distillation from DINOv2 preserves critical spatial semantics.
>
> **Q3: Clarification on Ablation (Teacher/Student Size).**
>
> **A3**:
>
> 1.**Why larger student (ConvNeXt) is worse? Overfitting.** In data-scarce regimes (10 demos), larger models optimize poorly compared to compact ones (ResNet-18), validating our choice of "compact student."
>
> 2.**Why teacher size has little impact? Information Saturation.** For the specific capacity of a ResNet-18 student, the knowledge in DINOv2-Small is already sufficient.
>
> **Q4: Did all baselines use the same Diffusion Policy head?**
>
> **A4**: Yes. All 2D baselines used the exact same Diffusion Policy head architecture and hyperparameters.
>
> **Q5: Why does $\pi_0$ fail (0% success) despite a better t-SNE score (0.296) than ResNet (0.087, 40% success)?**
>
> **A5**: A high clustering score only means the features are separated, not that they are useful.
>
> **$\pi_0$ (High score, 0% success)**: The model likely clusters frames based on task-irrelevant cues like background or lighting rather than the writing progress. While these clusters are tight, they map to incorrect "hovering" actions, causing the policy to get stuck consistently.
>
> **ResNet (Low score, 40% success)**: The low score confirms the features are messy and ambiguous, which often traps the policy in a loop of rewriting 'A'. However, Diffusion Policy is stochastic. In about 40% of trials, the noise sampling happens to produce a valid next-step action despite the noisy features, allowing the robot to "stumble" into the next stage. $\pi_0$, being confidently wrong, never gets this chance.
>
> *We have added these results and analyses to the revised paper.*

---

### Official Review · Reviewer_t9tz · 2025-11-03

**Soundness:** 3
**Presentation:** 3
**Contribution:** 2
**Rating:** 6
**Confidence:** 4

**Summary:**

The manuscript proposes the use of CNN-based vision backbones for visuomotor policies. It advocates for this solution based on the inductive biases that make CNNs easier to train in low-data regimes. To generate descriptive features, the authors propose the use of model distillation, distilling a large, generalist DINOv2 VIT into a ResNET 18 backbone using the non-robotic images in the ImageNET dataset.

Based on the proposed backbone architecture, a diffusion policy is implemented and evaluated in simulation and on real-world task. It is evaluated against different visual backbones, including transformers and large CNNs, and against pi0 as a generalist VLA policy. All models are trained on small demonstration datasets and results demonstrate the efficacy of the proposed approach. A further analysis of feature embeddings show the capability of the learned features to differentiate between different stages of robotic tasks.

**Strengths:**

- The approach shows strongly improved policy success rate with a simple and likely transferable approach.
- The resulting policy is evaluated in simulation and real-world experiments, and benchmarked against a range of visual backbones and policy types.
- The work approaches a robotic learning problem with only few demonstrations and no additional robotic data, which is a relevant practical scenario.

**Weaknesses:**

- The conceptual novelty of the work is limited, only applying a standard MSE distillation approach to DINO features.
- The evaluation of the work is limited to specialist visuomotor policies. Scaling of the approach to large data settings like VLAs or generalist diffusion policies like Octo [1] is not evaluated.
- The real-world evaluation settings only contain simple tabletop settings, more complex environments that test the generalization capability of the visual encoders with e.g. natural lighting or complex backgrounds are not contained in the test data.

[1] Octo Model Team, et al. "Octo: An open-source generalist robot policy." arXiv preprint arXiv:2405.12213 (2024).

**Questions:**

The work showcases an interesting observation in finding the right encoders for visuomotor policies. While the work is conceptually simple, it can provide an interesting stepping-stone toward this problem. However, to allow the reader to come to a valuable conclusion, the evaluation should demonstrate the limits of this approach and answer the following questions:
- How does the encoder scale with additional data? Does the encoder size correlate with the distillation data? Can a point be identified for VITs to be better?
- Can the approach profit from exposing the student model to large-scale robot data (OpenX Dataset)?
- Do the findings translate to generalist robot policies?

---

> ### Author Response · Authors · 2025-12-02
>
> We thank the reviewer for the positive assessment and for recognizing our work as a "relevant practical scenario" for robotic learning. We have conducted additional experiments to specifically address your questions regarding scaling limits and generalist policies.
>
> **Q1: How does the encoder scale with additional data? Can a point be identified for ViTs to be better?**
>
> **A1**: To answer this, we performed a Data Scaling Experiment on the real-world "Move Brush" task, training policies with dataset sizes ranging from $N=20$ to $100$.
>
> Table 1: Success Rates on "Move Brush" with Varying Demonstrations
> | # Demos | X-Distill (Ours) | ResNet-scratch | DINOv2 (ViT) | $\pi_0$ (VLA) |
> | :---: | :---: | :---: | :---: | :---: |
> | 20 | 58.3% | 0.0% | 0.0% | 41.7% |
> | 40 | 75.0% | 8.3% | 0.0% | 33.3% |
> | 60 | 83.3% | 25.0% | 0.0% | 66.7% |
> | 80 | 100.0% | 33.3% | 25.0% | 66.7% |
> | 100 | 100.0% | 58.3% | 41.7% | 58.3% |
>
> **Findings**:
>
> 1.**X-Distill saturates early**: Our method reaches 100% success with only 80 demos, showing extreme data efficiency.
>
> 2.**ViT crossover point**: DINOv2 (ViT) starts to improve at 100 demos (41.7%) but remains far behind. Extrapolating this trend, the "crossover point" where a ViT might outperform our distilled CNN likely requires thousands of trajectories. For the data-scarce regime (tens to hundreds of demos) targeted in this work, X-Distill is consistently superior.
>
> **Q2: Can the approach profit from exposing the student model to large-scale robot data (OpenX Dataset)?**
>
> **A2**: This is an interesting direction. In our current framework, we use ImageNet as the distillation corpus to ensure the encoder remains domain-agnostic and generalizable.
> While we did not train on OpenX from scratch due to computational constraints, we compared against $\pi_0$, a model explicitly pre-trained on large-scale OpenX data. As shown in the table above, even with OpenX pre-training, $\pi_0$ struggles to adapt to precise tasks with limited fine-tuning data (fluctuating between 33%-66%). This suggests that for data-efficient adaptation, our strategy of "Distilling ImageNet priors into a CNN" is currently more effective than "Pre-training on OpenX + Fine-tuning".
>
> **Q3: Do the findings translate to generalist robot policies?**
>
> **A3**: To test this, we evaluated $\pi_0$ (a state-of-the-art generalist VLA) on our tasks. The results (Table above and in the paper) show that generalist policies currently struggle with high-precision manipulation (e.g., failing completely on the "Writing" task) when data is limited.
>
> Therefore, our findings are most relevant to Specialist Policies where precision and data efficiency are paramount. However, we believe the core insight—that compact architectures with distilled priors adapt faster—could be valuable for designing the adapter layers or lightweight vision heads of future generalist policies.

---

### Meta-Review · Area_Chair_1q31 · 2026-01-06

**Summary:**

Reviewers acknowledge the strong empirical results of X-Distill in extremely low-data robotic learning, especially its real-world efficacy and simplicity. However, major concerns center on limited evaluation scope (e.g., narrow real-world settings), limited novelty (e.g., standard MSE loss without architectural or algorithmic innovation), and insufficient depth in comparisons to prior work and follow-up investigations.

**Reviewer Concerns:**

- Several requests on more evaluation results have been addressed, including: scaling with more data (up to 100), comparison to R3M, comparison to LoRA.
- Some clarifications, e.g, policy head uniformity, t-SNE explanations, are resolved.
- Real-world complexity is not addressed. The simplicity of the task conditions may cause the discrepancy of the results.
- How the proposed method integrates with other policy networks/VLAs is not addressed.
- Explanations for counterintuitive results (e.g., larger teacher no gain, LoRA underperformance) remain post-hoc and lack ablation or diagnostic probing.
- The novelty issue is not addressed.

**Reviewer Scores:**

- t9tz: Likely unchanged or slightly lowered. Key concerns—novelty, real-world complexity, and generalist applicability—remain unaddressed.
- JJPk: Likely unchanged. While requested baselines were added, the lack of deeper validation limits confidence in conclusions.
- hLzF: Likely unchanged. Similarly, some new experimental results are provided while the rebuttal offers no analysis of e.g., why distillation wins.

---

### Decision · Program_Chairs · 2026-01-26

Reject